# Factors controlling the productivity of tropical Andean forests: Climate and soil are more important than tree diversity

Jürgen Homeier[1,2], Christoph Leuschner[1,2]

[1]Plant Ecology and Ecosystems Research, University of Goettingen, Untere Karspüle 2, 37073 Goettingen, Germany
[2]Centre for Biodiversity and Sustainable Land Use, University of Goettingen, Untere Karspüle 2, 37073 Goettingen, Germany

*Correspondence to*: Jürgen Homeier (jhomeie@gwdg.de)

**Abstract.** Theory predicts positive effects of species richness on the productivity of plant communities through complementary resource use and facilitative interactions between species. Results from manipulative experiments with tropical tree species
indicate a positive diversity-productivity relationship (DPR), but the existing evidence from natural forests is scarce and contradictory. We studied forest aboveground productivity in more than 80 humid tropical montane old-growth forests in two highly diverse Andean regions with large geological and topographic heterogeneity, and related productivity to tree diversity and climatic, edaphic and stand structural factors with likely influence on productivity. Main determinants of wood production in the perhumid study regions were elevation (as a proxy of temperature), soil nutrient (N, P and base cation) availability, and
forest structural parameters (wood specific gravity, aboveground biomass). Tree diversity had only a small positive influence on productivity, even though tree species numbers varied largely (6-27 species per 0.04 ha). We conclude that the productivity of highly diverse Neotropical montane forests is primarily controlled by thermal and edaphic factors and stand structural properties, while tree diversity is of minor importance.

## 1 Introduction

Research into the diversity-productivity relationship (DPR) has recently shifted its focus on forests because of their importance for the global carbon cycle and the high biodiversity especially of tropical forests (Mori et al., 2017; Fei et al., 2018). Results from mixed species plantations with tropical trees indicate a positive DPR in the majority of cases (Sapijanskas et al. 2014; Huang et al., 2018; Schnabel et al., 2019), suggesting complementary resource use and facilitative interactions between different tree species (McIntire et al., 2014; Chen et al., 2016) The results of these experiments may provide valuable insights
into the mechanisms of diversity effects on productivity (Tuck et al., 2016; Schnabel et al., 2019), but they are difficult to extrapolate to tropical old-growth forests due to the young age and cohort-structure of these tree plantations. Observational studies on the linkage between tree diversity and productivity in natural or near-natural tropical forests are usually burdened with additional confounding factors and thus are more difficult to interpret than experiments, but they have the advantage of considering natural systems. The few existing observational studies on the DPR in tropical forests have produced contradictory
evidence. On Mt. Kinabalu, Malaysia, aboveground NPP was higher in more diverse tropical montane forest plots (Aiba et al.,

2005), but no effects of species richness on productivity were found in logged tropical lowland forests in Guyana (van der Sande et al., 2018). Chisholm et al. (2013) analyzed the tree diversity-productivity relation in 25 permanent forest plots from temperate to tropical regions and found a positive diversity effect when plot size was small (0.04 ha), but the effect disappeared at larger spatial scales (0.25 and 1 ha). In a global analysis of forest productivity data, Liang et al. (2016) found that positive DPRs predominate, but negative relations do also exist, though restricted to a few of the tested data sets.

The existing observational data from tropical forests suggest that (i) tree diversity may enhance productivity under certain conditions, and (ii) a positive DPR might be more prominent in small plots. To date, it is not clear under which abiotic conditions and stand structural settings a positive DPR does emerge, and how important tree diversity is compared to abiotic and other biotic determinants of productivity.

The search for abiotic determinants of tropical forest productivity has mainly concentrated on soil nutrient availability, solar irradiance, and precipitation, revealing clear dependencies in studies along environmental gradients (Schuur, 2003; Girardin et al., 2014; Malhi et al., 2017). Many tropical forests, especially those in the lowlands, seem to be P limited (Aragao et al. 2009, Quesada et al. 2012), while N limitation may be more influential in tropical montane forests (TMF) (Tanner et al. 1998; Moser et al. 2008, Homeier et al. 2012, Fisher et al. 2013). Fertilization experiments and studies on foliar nutrient levels across environmental gradients suggest that the net primary productivity (NPP) of tropical forests tends to increase with soil fertility, notably N and P availability, but locally also with base cation supply (Kitayama & Aiba, 2002; Cleveland et al.; 2011; Malhi et al., 2004; Homeier et al., 2010; Unger et al., 2012; Fisher et al., 2013; Banin et al., 2014; Hofhansel et al., 2015). Bruijnzeel & Veneklaas (1998) assumed that the productivity of TMFs is primarily limited by cloudiness, i.e. low solar radiation, which is supported by a modeling study (Fyllas et al., 2017). Precipitation should play a minor role in TMFs (Zimmermann et al., 2010). Less clear is the role of temperature. The elevational temperature decrease can act on plants in multiple ways, directly through a limitation of photosynthetic activity, and of carbon, water and nutrient cycling in the plant, and it may influence stem, leaf and root morphology. The impairment of nutrient supply by low soil temperatures is an indirect effect. Yet, plants can adapt to lower temperatures, thereby partly reducing these limitations (Lambers & Oliveira, 2019). The majority of studies along elevation transects found a NPP decline with decreasing temperature in tropical mountains (Kitayama & Aiba, 2002; Girardin et al., 2010; Cleveland et al.; 2011; Leuschner et al., 2013; van de Weg et al., 2014). However, an analysis of a global database failed to show a temperature dependence of NPP in tropical forests (Luyssaert et al., 2007). This fits to modeling results suggesting that trait variation with elevational species turnover in TMFs may partly capture the effect of temperature on tree metabolism and growth (Fyllas et al., 2017).

Biotic controls related to stand structural properties, notably stem density, basal area and wood density, and the functional composition of the community can also affect forest productivity. Some studies found that forest NPP increased with standing aboveground biomass (AGB) (Keeling & Phillips, 2007; Pan et a., 2013; Lohbeck et al., 2015), but the relation does not seem to be very close, since woody tissue residence time, i.e. the ratio between AGB and aboveground productivity, varies considerably with soil fertility and climate in tropical moist forests (Malhi, 2012; Quesada et al., 2012; Chisholm et al., 2013; Malhi et al., 2017). Variation in leaf area index and leaf functional traits such as LMA (leaf mass per area) and foliar N and P

content should be major determinants of net primary productivity (van de Weg et al., 2014; Wittich et al., 2012; Finegan et al., 2015), as they influence the photosynthetic capacity of the canopy and radiation interception. Indeed, the modeling study of Fyllas et al. (2017) indicated for tropical forests along an Andean elevation gradient that the influence of LMA, foliar N and P content and wood density of the species was a more important determinant of NPP than temperature, which is in line with the findings of Luyssaert et al. (2007). From the partly contradicting results we conclude that a multi-factorial approach is

needed to understand the determinants of tropical forest productivity. Yet, most of the existing studies have investigated only one or two of the mentioned factors.

    Tropical montane forests cover all humid tropical mountains between ~1000 and 3000 m asl with widest distribution in the Andes and Central American mountains, and more restricted occurrence in the mountains of the Paleotropis. They differ from tropical lowland forest in terms of lower canopy heights, higher stem densities and lower AGB. TMFs play important

roles in the provision of water for human populations, as stores of carbon, and tropical mountains are known as hotspots of biodiversity (Ashton, 2003; Bruijnzeel et al., 2010; Spracklen & Righelato, 2014; Fahey et al., 2015; Rahbeck et al., 2019). TMFs occur along steep thermal and edaphic gradients, making them attractive for the study of abiotic controls of tropical forest productivity and the role of tree diversity. In this study in two Andean elevation transects, we combine biomass, productivity and tree diversity data with comprehensive soil chemical data to identify the most important abiotic and biotic

drivers of forest productivity in Neotropical montane forests (Figure 1). In permanent plots in old-growth forests, we measured coarse wood production (WP) and fine litter production (only one of the transects) and related these components of aboveground primary production to the availability of the five macronutrients N, P, Ca, K and Mg in the soil, to elevation as a proxy of temperature, and to stand structural properties and tree diversity in the plots. We chose a plot size of 0.04 ha (20 m × 20 m) in order to meet plot homogeneity requirements in the rugged terrain. Moreover, assumed diversity effects are more

likely to emerge at small spatial scales (Chisholm et al., 2013). This allowed us to investigate a larger number of plots across extended environmental gradients (2000 m of elevation distance or ~10 K variation in mean annual temperature; large bedrock, topographic and soil variation) and to encounter considerable variation in tree diversity (6 to 27 species per 0.04 ha). The analyses were restricted to old-growth forests with closed canopy in order to exclude variation in stand structure and species composition related to successional processes. We employed structural equation modeling for exploring mutual inter-

relationships between the likely abiotic and biotic drivers of productivity.

    Based on the assumption that precipitation is not a growth-limiting factor in this humid to perhumid study region, we hypothesized in accordance with the existing literature that (i) temperature is the principal productivity-determining factor in the study regions, (ii) soil nutrient availability is another, but secondary, influential abiotic factor acting in TMFs mainly through N availability, and (iii) tree diversity is a less important driver of productivity than abiotic factors.

 **2 Methods**

**2.1 Study transects**

The study was conducted in permanent plots in old-growth montane forests established along two elevation transects on the eastern slope of the Ecuadorian Andes in the provinces Napo (Napo transect) and Loja and Zamora-Chinchipe (Loja transect) (Figure A1). The chosen plot area was 20 m x 20 m (400 m²), a size that has previously been used in surveys of tropical forest

diversity, because the area is large enough to give representative information on stand structure and species composition, while the patch size is in tropical montane old-growth forests small enough to guarantee sufficient stand structural and edaphic homogeneity on the plot level. All plots were placed in forest patches without signs of recent disturbance and in sufficient distance to special microhabitats such as ravines and ridges. By selecting small-sized plots solely in old-growth portions of the forest, we attempted to reduce the variation in structural and biological parameters resulting from the mosaic of different

successional stages typically existing in natural forests.

The climate in both transect regions is perhumid throughout the year with a mean annual precipitation >2000 mm at all elevations and lack of a distinct dry season (Bendix et al., 2008; Salazar et al., 2015). All study sites are characterized by aridity index values >0.74 after Trabucco et al. (2009).

Soil properties vary with elevation and differ between the two transects. The Loja transect is characterized by relatively

nutrient-poor soils mostly on metamorphic schists and sandstones with a slightly better nutrient supply at lower elevations and in valleys and less favorable supply of N and P on upper slopes and at higher elevations (Wilcke et al., 2008; Werner & Homeier, 2015). In contrast, the Napo transect has more fertile soils, which developed on volcanic deposits or limestone. In this transect, N mineralization rate decreases with elevation, while plant-available P increases (Unger et al., 2010). The availability of the five plant macronutrients N, P, Ca, K and Mg was analyzed in the soil of all study plots together with soil

acidity in order to cover the physiologically most meaningful soil chemical properties. For characterizing N availability, incubation experiments to determine the net release of $NH_4^+$ and $NO_3^-$ through mineralization and nitrification ($N_{min}$) were conducted in the topsoil. In addition, the bulk soil C/N ratio was determined in the topsoil for characterizing the decomposability of soil organic matter and thus give an independent measure of potential N supply (Pastor et al., 1984). The increase in organic layer depth with elevation in both transects, accompanied by greater C/N ratios and lower pH, is probably

related to higher soil organic matter contents and indicates in our study areas a decrease in nutrient availability as a result of reduced organic matter turnover at lower temperatures. P availability was estimated as resin-exchangeable P, i.e. the exchange of phosphate ions by anion exchange resins, which may give a minimum estimate of plant-available P ($P_{av}$). The plant availability of Ca, K and Mg ($CaKMg_{ex}$), and also of potentially toxic $Al^{3+}$, was quantified by salt exchange (0.2 N $BaCl_2$ solution), applying a standard protocol for the chemical analysis of forest soils (for analytical details see Unger et al. (2010)

and Wolf et al. (2011)). Soil pH was measured in soil suspended in 1 M KCl. Since most fine roots are concentrated in the topsoil (Soethe et al., 2006), where also the vegetation effect on the soil is largest, the statistical analyses focus on the chemistry of the 0-10 cm layer and its importance for the vegetation (see Supplement S2 for the soil chemical properties of the plots).

## 2.2 Plot inventory and forest productivity

The Loja transect consists of 54 plots that are equally spread over three chosen elevation levels (18 plots each at ~1000, ~2000

and ~3000 m a.s.l.) (see Supplement S1 for the stand properties of the plots in both transects). From the Napo transect, we
included data from 66 plots that were located in a comparable elevation range (960 – 3100 m asl). In both transects, all trees
with diameter at breast height (dbh) ≥10 cm in the 400 m$^2$–plot were recorded with their dbh and identified to species. From
unknown tree species, specimens were collected for later identification at Ecuadorian herbaria (HUTPL, LOJA, QCA, QCNE)
or by international specialists for difficult groups.

Data on AGB, annual AGB increment (coarse wood production, WP) and WSG (wood specific gravity) of all trees ≥10 cm
dbh in the plots were available from earlier studies (Loja transect: Wolf et al. (2011), Leuschner et al. (2013); Napo transect:
Unger et al. (2012), Kessler et al. (2014)). AGB and WP of all trees ≥10 cm dbh in a plot were estimated as the sum of the
biomass of all tree individuals and the corresponding increment of these trees in a second stem diameter inventory after 1-5
years. For quantifying AGB, we applied the allometric equation of Chave et al. (2005) for tropical wet forests. WP was

calculated as the difference between the two AGB estimates on an annual basis. For the Loja transect we additionally calculated
net aboveground productivity (NPP$_a$) as the sum of WP and annual fine litter production that was recorded in litter traps (data
available from Wallis et al., 2019).

Leaf area index (LAI) was estimated by synchronous measurement with two LAI 2000 Plant Canopy Analyzers (LI-COR Inc.,
Lincoln, NE, USA) that were operated in the remote mode during periods of overcast sky. Simultaneous readings were taken

below the canopy at 2 m height above the ground and in nearby open areas ("above-canopy" reading) to record incoming
radiation. Data for the Napo transect were taken from Unger et al. (2013). For the Loja transect, the LAI measurements were
conducted in 2011, adopting the methods described in detail in Unger et al. (2013).

## 2.3 Data analysis

Tree diversity was calculated according to the individual-based rarefaction method (Gotelli & Colwell, 2001) as the number

of species (stems with dbh ≥10 cm) expected in a random sample of 14 trees in a 400 m²-plot (14 being the smallest number
of tree individuals recorded in our plots). Linear regression analyses were applied to identify significant relationships between
AGB, WP, tree diversity, LAI, stem density, WSG and soil properties as dependent variables, and elevation (as a proxy of
temperature) as independent variable.

We used principal components analysis (PCA) to reduce the number of soil variables and to ensure that the subsequent analyses

were not affected by the problem of multi-collinearity. We conducted two PCAs using the R package 'pcaMethods', one with
the merged data from both transects, and another one for the Loja transect. The following ten soil variables were included in
the PCAs: organic layer depth, mineral soil pH (KCl), exchangeable K, Mg, Ca, and Al contents, resin-exchangeable P content,
C/N ratio, topsoil N mineralization and nitrification rate. The plot scores on the first two resulting principal components (PC
1 and PC 2) were included in the subsequent analyses.

Structural equation modeling was used for identifying the direct influence of abiotic factors (elevation, soil PCs) on forest stand properties and tree diversity and to assess the direct and indirect influences of these factors on productivity. For each initial structural equation model (SEM), we included the two principal soil components (PC1 and PC2) from the respective PCA, and the factors elevation, WSG, and tree diversity to predict AGB and stand productivity (wood production WP or $NPP_a$). WSG was selected from the stand properties, as it showed a stronger correlation to stand productivity than the other measured
variables (stem density and LAI).

    Based on the existing knowledge about the dependency of productivity and AGB on environmental and stand properties, and assumed diversity effects on productivity, we developed an initial model of interaction paths between the environmental parameters (elevation, soil PC 1 and PC 2), WSG and tree diversity, and the target variables (AGB and WG or $NPP_a$) (Figure 1). We then fitted two different SEMs, one for the complete set of 83 plots from both transects (where all information was
available: 54 Loja plots and 29 Napo plots), and another one for the Loja transect (54 plots), where $NPP_a$ (and not only WP) data are available.

    IBM SPSS AMOS 24 software (Arbuckle 2016) was used to fit the normalized data to the hypothesized path model and to determine path coefficients using the maximum likelihood method. We assessed the goodness of model fit using the $\chi^2$ value, the associated p value, AIC, RMSEA (root mean square error of approximation), and CFI (comparative fit index). Since we
used SEM in an explorative way, the original model has been subject to modification. We iteratively removed insignificant paths to test whether incorporation of those paths in the model significantly increased the $\chi^2$ value and CFI, and reduced the AIC and the RMSEA of the model. For all dependent variables, we calculated $R^2$ values that indicate the proportion of variance explained by the model.

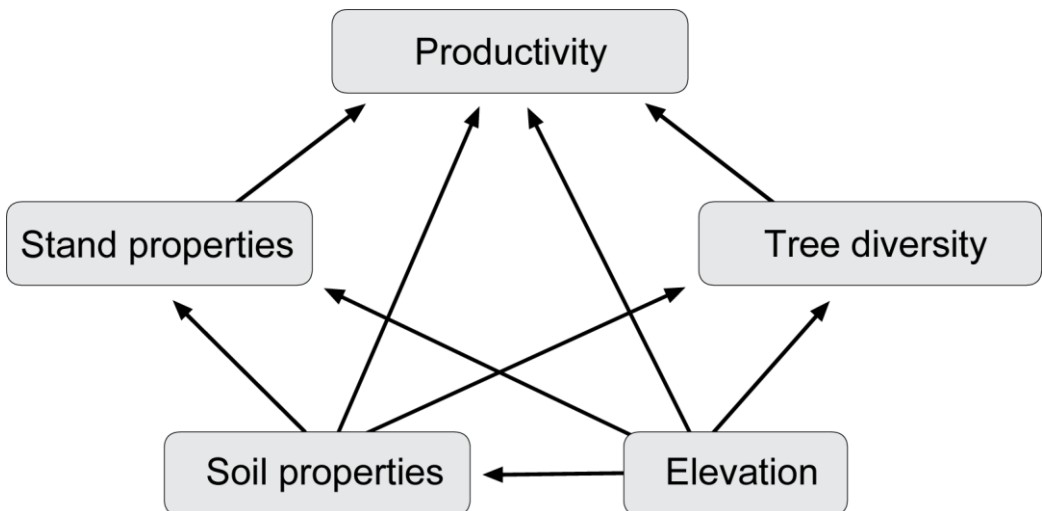


**Figure 1.** Conceptual model of plausible interaction pathways between environment (elevation, soil) and forest structure as drivers of productivity. Arrows drawn from the two boxes Forest stand properties and Soil properties indicate that pathways from the respective variables (Forest stand properties: wood specific gravity, stem density, LAI; Soil properties: first two PCA axes) were included in the models.

**3 Results**

 **3.1 Patterns of tree diversity and productivity**

The two transects differed considerably with respect to floristic composition. Most important tree families in the Loja transect (in order of abundance) were Lauraceae, Melastomataceae, Cunoniaceae, Moraceae and Clusiaceae, in the Napo transect Euphorbiaceae, Fabaceae, Lauraceae, Moraceae and Rubiaceae. The number of stems per plot varied between 14 and 61 and the number of tree species between 6 and 27 (Supplement S1). Tree species richness showed no significant decline with

elevation (Figure 2a). AGB decreased on average by 50-90 Mg ha$^{-1}$ per km elevation in both transects (Figure 2b). Biomass was between 200 Mg ha$^{-1}$ (lower elevations) and 100 Mg ha$^{-1}$ (upper elevations) higher in the Napo transect on more fertile soil than in the Loja transect.

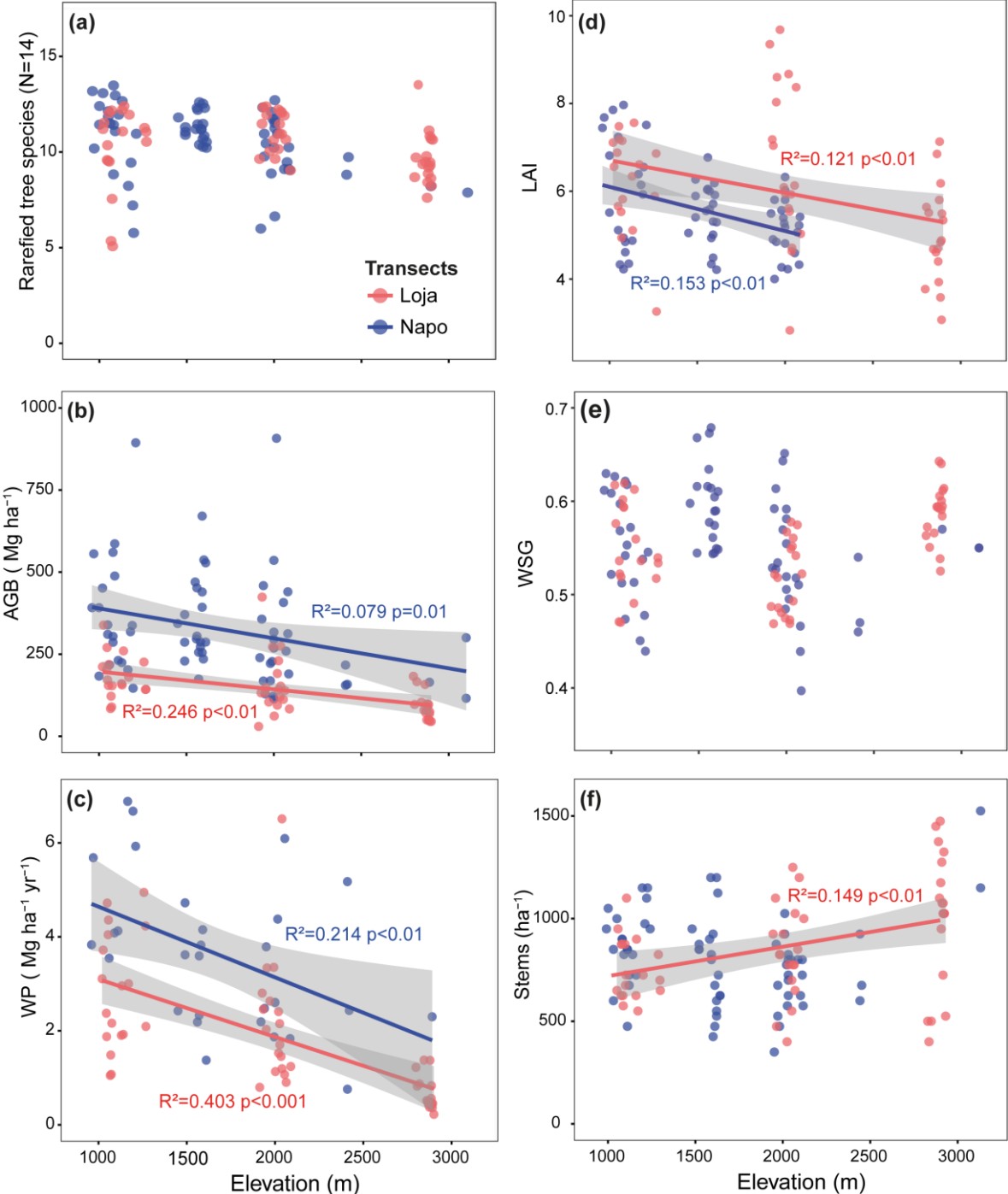

**Figure 2.** Variation of tree diversity (a), aboveground biomass (b), wood production (c), LAI (d), wood specific gravity (e), and stem density
(f) along the two studied forest transects (red dots: Loja-transect, blue dots: Napo-transect). The number of plots available for a given
parameter and transect varies, as not all parameters were measured in all plots (a: Loja 54 plots / Napo 64 plots; b: 54/66; c: 54/29; d: 53/60;
e: 54/66; f: 54/66). Data presented in b, c, e and f were compiled from Unger et al. (2012), Leuschner et al. (2013), Kessler et al. (2014);
data from the Napo transect in d are from Unger et al. (2013).

The AGB decrease was linked to a decline in WP between 1000 and 3000 m by about 1.3 - 1.5 Mg ha$^{-1}$ yr$^{-1}$ per km elevation with a generally lower productivity (by about 1.0 - 1.5 Mg ha$^{-1}$ yr$^{-1}$) in the Loja transect compared to the Napo transect (Figure 2c). LAI decreased with elevation in both transects by about 1 m$^2$ m$^{-2}$ per km elevation with slightly higher LAI values in the Loja transect (Figure 2d). WSG showed no systematic variation with elevation (Figure 2e). The expected stem density increase with elevation was only observed in the Loja transect (Figure 2f).

### 3.2 Identification of principal soil factors

In the two data sets (both transects merged; Loja transect only), the first two PCA axes explained 63 % and 61 %, respectively, of the variation in soil factors across all plots. In the merged data set (Figure 3, Figure B1 and Table B1), the first axis represented mainly the basic cations Mg, Ca and K, soil pH and organic layer depth, whereas the second axis stood for N supply and P$_{av}$. In the PCA of the Loja transect (Figure B2 and Table B2), soil parameters were represented similarly, but P$_{av}$ had a higher loading on the first axis. In general, the soils of the Napo transect were characterized by higher fertility, as indicated by on average markedly higher topsoil net N mineralization rates and higher P availability in the mineral topsoil (see Supplement S2).

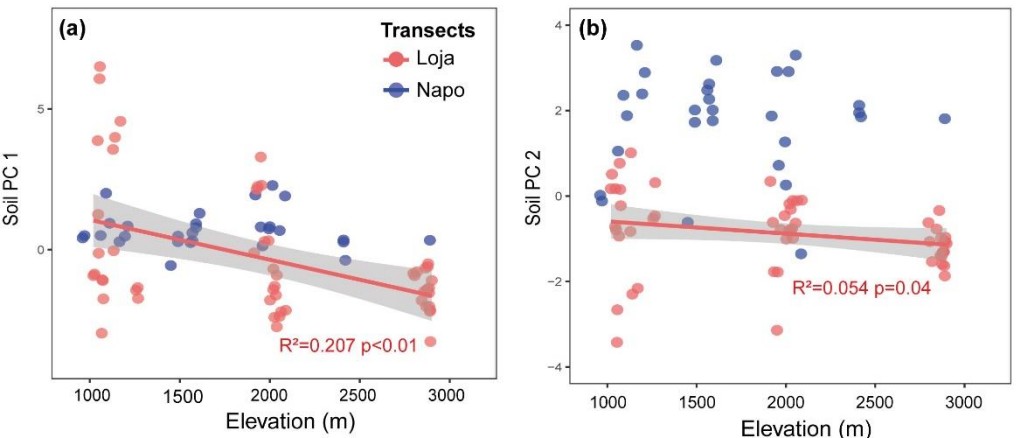

**Figure 3.** Variation of soil properties along the two forest transects (red dots: Loja-transect, blue dots: Napo-transect). Shown are the (a) first and (b) second principal components of the soil PCA (see Figure B1 and table B1).

### 3.3 Factors determining wood production in the merged data set

In the merged data set (83 plots; Figure 4, Table 1), aboveground tree biomass (AGB) was positively influenced by soil fertility (standardized total effects of PC1 and PC2, 0.30 and 0.45, respectively) and tree diversity (0.23). The three most important determinants of WP were elevation (negative influence, standardized direct effect -0.58), soil PC2 (0.29) and WSG (-0.25).

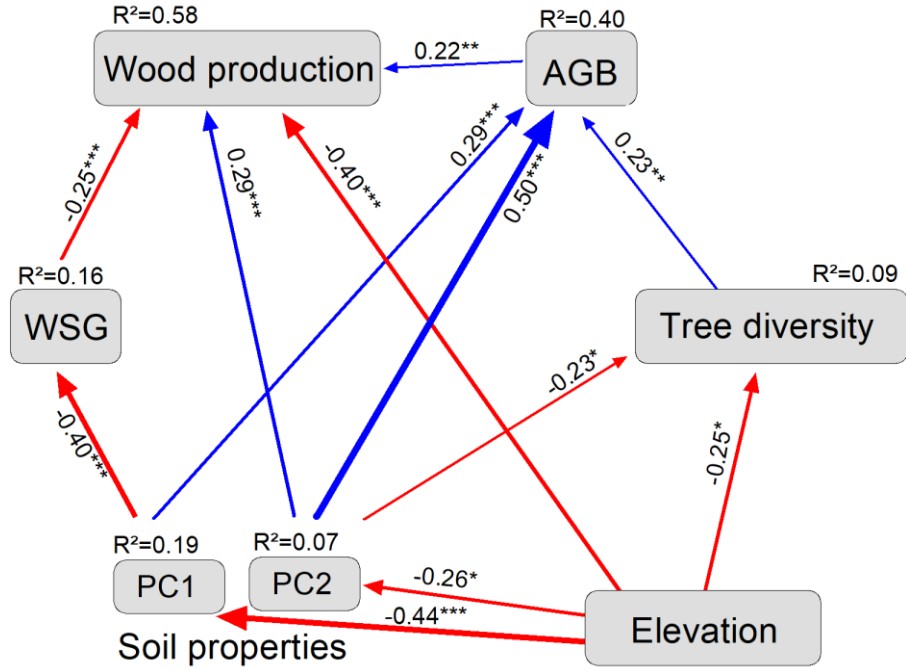

**Figure 4.** Final model for wood production (Loja-transect and Napo-transect: 83 plots). Structural equation model (chi-square = 8.4, 9 df, p = 0.50, AIC 60.4, RMSEA < 0.01, CFI = 1.00) with standardized path coefficients. The size of the arrows is proportional to the strength of the paths, their significance is indicated by asterisks (* $P < 0.05$, **$P < 0.01$, *** $P < 0.001$), blue arrows indicate positive and red negative estimates. AGB: aboveground biomass; WSG: wood specific gravity; PC1, PC2: first two axes of the soil PCA (Figure B1 and Table B1).

**Table 1.** Standardized direct, indirect and total effects of elevation, various stand structural and soil chemical parameters on coarse wood production (WP) and aboveground biomass (AGB) according to the SEM analysis in Figure 4. Standardized path coefficients are shown.

| Factors | Direct | Indirect | Total |
|---|---|---|---|
| **WP** | | | |
| Elevation | -0.397 | -0.183 | -0.580 |
| PC1 | | 0.163 | 0.163 |
| PC2 | 0.285 | 0.099 | 0.385 |
| AGB | 0.222 | | 0.222 |
| WSG | -0.245 | | -0.245 |
| Tree diversity | | 0.051 | 0.051 |
| **AGB** | | | |
| Elevation | | -0.302 | -0.302 |
| PC1 | 0.295 | | 0.295 |
| PC2 | 0.447 | | 0.447 |
| Tree diversity | 0.230 | | 0.230 |

The direct negative effect of elevation on WP was prominent and this influence was enhanced through indirect relationships via negative effects of elevation on soil properties and tree diversity (standardized indirect effect -0.18). However, soil factors were also important drivers of wood production, either directly (in case of PC2, which was related to N mineralization, 0.29), or indirectly through a soil N effect (PC2) on AGB and diversity (together 0.10), and effects of base cation availability and pH (PC1) on WSG and AGB. Tree diversity itself had only a relatively weak indirect positive effect on WP through its influence on AGB (standardized total effect 0.05).

### 3.4 Factors determining NPP$_a$ in the Loja transect

The model for aboveground net primary production (i.e. WP plus fine litter production), built only with the Loja transect data, identified elevation and WSG as the main direct determinants of NPP$_a$ (both effects negative; standardized direct effects of -0.67 and -0.39, respectively; Figure 5, Table 2). Soil properties were also influential (mainly PC1 with N availability), but only indirectly through a negative effect on WSG and a positive effect on diversity (standardized total effect 0.30). Tree diversity had a significant positive, but relatively weak direct effect on NPP$_a$ (0.18), which was enhanced by an indirect influence via AGB (standardized total effect 0.27).

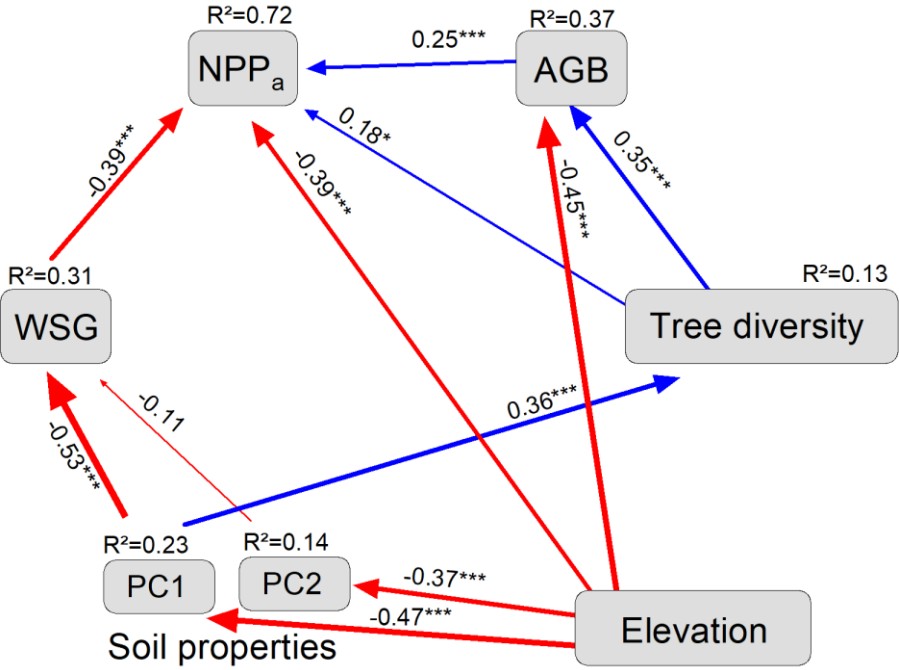

**Figure 5.** Final model for NPP$_a$ (Loja-transect: 54 plots). Structural equation model (chi-square = 9.9, 10 df, p = 0.45, AIC 59.9, RMSEA < 0.01, CFI = 0.99) with standardized path coefficients. The size of the arrows is proportional to the strength of the paths, their significance is indicated by asterisks (* P < 0.05, **P < 0.01, *** P < 0.001), blue arrows indicate positive and red negative estimates. AGB: aboveground biomass; WSG: wood specific gravity; PC1, PC2: first two axes of the soil PCA (see Figure B2 and Table B2).

**Table 2.** Standardized direct, indirect and total effects of elevation, various stand structural and soil chemical parameters on NPP$_a$ and aboveground biomass (AGB) (Loja transect) according to the SEM analysis in Figure 5. Standardized path coefficients are shown.

| Factors | Direct | Indirect | Total |
|---|---|---|---|
| **NPP$_a$** | | | |
| Elevation | -0.393 | -0.272 | -0.665 |
| PC1 | | 0.304 | 0.304 |
| PC2 | | 0.042 | 0.042 |
| AGB | 0.253 | | 0.253 |
| WSG | -0.391 | | -0.391 |
| Tree diversity | 0.178 | 0.088 | 0.266 |
| **AGB** | | | |
| Elevation | -0.447 | -0.60 | -0.506 |
| PC1 | | 0.126 | 0.126 |
| PC2 | | | |
| Tree diversity | 0.346 | | 0.346 |

## 4 Discussion

In a global perspective, forest productivity increases with temperature in cooler regions, and with precipitation in drier regions (Reich & Bolstad 2001; Schuur 2003). In the absence of temperature and precipitation measurements and soil moisture data from the plots, we used elevation as a proxy for the thermal conditions. We further assumed that soil moisture is rarely constraining forest productivity in both study regions, as precipitation totals range well above 2000 mm yr$^{-1}$ throughout the Napo and Loja transects (Bendix et al. 2008; Salazar et al., 2015) and soil moisture is usually high in Andean TMFs

(Zimmermann et al., 2010). Our assumption is supported by the highly significant negative influence of elevation on WP, suggesting that the temperature decrease with elevation was physiologically much more important than a possible water shortage effect on productivity. Both the SEMs and correlation analyses show that tropical forest productivity largely decreases (by 1.3 - 1.5 Mg ha$^{-1}$ yr$^{-1}$ km$^{-1}$) with a concurrent decrease in mean annual temperature from ~20 to ~10 °C in the two transects, confirming earlier observations (Aiba et al., 2005; Cleveland et al., 2011). Reduced temperatures may negatively affect tree

growth through a variety of direct and indirect influences on carbon gain, carbon allocation, and nutrient acquisition. Photosynthesis measurements revealed by 36 % and 18 % lower community means of light-saturated net photosynthesis rates (A$_{max}$) at 3000 m compared to the community means at 2000 m and 1000 m, respectively (Wittich et al., 2012). Reduced canopy carbon gain, in combination with the elevational LAI decrease by 1 m$^2$ m$^{-2}$ km$^{-1}$, might largely explain the productivity decline from 1000 to 3000 m in the Ecuadorian Andes (Leuschner et al., 2013). In apparent contradiction, the tree-based carbon

balance model of Fyllas et al. (2017) suggests for the Peruvian Andes that the apparent temperature effect on photosynthesis

and productivity in the 3300 m-elevation gradient is manifested through leaf trait variation associated with the elevational species turnover rather than through a temperature dependency of photosynthesis itself. In fact, light-saturated net photosynthesis ($A_{max}$) did not change between 250 and 3500 m asl in this transect (Fyllas et al., 2017). These results contradict a pantropical analysis (c. 170 tree species, 18 sites) showing that $A_{max}$ decreases on average by 1.3 µmol m$^{-2}$ s$^{-1}$ per km elevation in tropical mountains (Wittich et al., 2012). We assume that local differences in soil fertility and climatic factors such as cloudiness or very high precipitation, causing temporal soil anoxia, may have a large influence on elevational patterns in foliar nutrient content and photosynthetic capacity. Moreover, the $A_{max}$ study of Wittich et al. (2012) with 40 Andean tree species from 21 families points at a considerable influence of phylogeny on photosynthetic capacity. Species turnover between elevations and also between different locations might therefore explain some of the differences in photosynthetic carbon gain and productivity, independently of temperature differences.

Low-temperature effects on nutrient supply and acquisition processes, which enhance nutrient deficiency, could well be additional factors causing productivity reductions and increased belowground C allocation at higher elevations (Graefe et al., 2008; Moser et al. 2011; Leuschner et al., 2013). Indeed, pronounced leaf trait changes were observed in the Loja transect from larger, thinner and N-richer leaves at 1000 m to smaller, thicker and N-poorer leaves at 3000 m (Homeier et al., unpubl.), suggesting that temperature may act on the forest carbon cycle partly through N and P availability as an environmental filter, sorting tree species according to their leaf traits from more acquisitive to more conservative at higher elevations.

Temporal water limitation, if it were relevant in the two regions, should be more prominent at lower elevations with lower rainfall, which should lead to a positive, and not a negative, elevation effect on productivity. We cannot exclude, however, that high soil moisture in combination with temporal hypoxia in the rhizosphere of the high-elevation stands is contributing to the productivity decline toward the timberline (Moser et al., 2011).

Other climatic factors with potential impact on productivity such as solar radiation, air humidity and leaf wetness were not considered in our analysis due to lack of suitable local data. Solar radiation typically reaches a minimum at montane or upper montane elevation due to cloud immersion, which can reduce carbon gain (Malhi et al., 2017). Including these factors in the analysis may lead to somewhat different conclusions on the key determinants of forest productivity (Finegan et al., 2014; Fyllas et al., 2017; Malhi et al. 2017).

Our study in two regions of the northern Andes is the first worldwide to analyze the dependence of tropical montane forest production on the availability of all five quantitatively most important plant nutrients (N, P, Ca, K, Mg). P availability has been reported to limit or co-limit forest productivity at many tropical lowland sites (Tanner et al., 1998; Paoli & Curran, 2007; Cleveland et al., 2011; Wright et al., 2011; Dalling et al., 2016b), whereas N seems to limit tropical forest productivity more often at higher elevations (Tanner et al., 1998; Benner et al., 2010; Wolf et al., 2011; Dalling et al., 2016a). The prominent direct and indirect effects of PC1 and PC2 on productivity suggest that all three nutrient components (N, P and base cations) play roles in the limitation of forest productivity in the study regions.

A high net N mineralization rate positively influenced WP in the merged dataset, but the indirect effect through AGB dominated. The large difference in N mineralization rates between the two transects with high values on the predominantly

volcanic soils in the Napo region, but low values on the non-volcanic soils of the Loja region, probably explains also the considerably higher AGB and consequently WP in the Napo transect as compared to the Loja transect (Figure 2). The trees in the less fertile Loja transect most likely allocate a higher proportion of their photosynthates to root production and to root symbionts (Vicca et al., 2012; Doughty et al., 2017), which must reduce wood production. The availability of base cations and soil acidity influenced productivity in both models only indirectly, mainly through an effect on WSG (higher wood density on

acidic, base-poor soils).

The PCAs also showed that the three main nutrient categories (P, N, base cations) varied more or less independently from each other across the plots. Moreover, their relative importance varied between the Loja and Napo data sets, suggesting that soil nutrient limitation of forest productivity is depending more on region, bedrock type and soil age than on elevation. The different paths from soil nutrients to productivity and the weak correlations among $P_{av}$, $N_{min}$ and base cations across our data set imply

that no single nutrient element can serve as a good indicator of overall nutrient availability or soil fertility in the study regions. This is also valid for pH, which showed the tightest relation to $CaKMg_{ex}$ (Loja transect) or organic layer depth (merged transects), but weaker or no association to $P_{av}$, $N_{min}$ and soil C/N ratio. Soil fertility assessments in Andean montane forests should consider N, P and also basic cations, when forest productivity shall be related to soil fertility. The widely held assumption that net primary production in tropical lowland forests is primarily limited by low P availability, and TMF

productivity predominantly by low N availability (Vitousek & Sanford, 1986; Tanner et al., 1998; Fisher et al., 2013), may thus require adaptation.

Various studies have identified WSG as a good predictor of the diameter increment of tropical trees, and high WSG has been linked to low aboveground productivity in tropical lowland forests (Malhi et al., 2004; Poorter et al., 2008; Finegan et al., 2015). In the Ecuadorian Andes, the negative effect of average WSG on productivity was as strong as the positive biomass

(AGB) influence, suggesting that negative edaphic effects on productivity, notably high soil acidity and low availability of basic cations, are exerted in part indirectly through an increase in wood specific gravity (Unger et al., 2012). A recent study in South-east Asian tropical forests showed that WSG does neither have a direct effect on biomass production (Kotowska et al., 2021) nor on tree water consumption, even though harder wood tends to be associated with lower growth rates (Muller-Landau, 2004; Hoeber et al., 2014). Yet, WSG is known to be associated with most structural and functional wood properties (Chave

et al., 2009), and it may also be related to anatomical attributes such as pit membrane characteristics that influence sap flux density, which itself is positively related to productivity (Kotowska et al., 2020). In addition, our data support a causal link from higher N availability through lower WSG to higher productivity, which is suggested by N fertilization experiments in conifers (Jozsa & Brix, 1989) and a negative correlation between wood density and soil fertility in continent-wide wood property analyses (Chave et al., 2009). N fertilization experiments in Andean TMFs support the positive effect of N on

productivity (Homeier et al., 2012; Fisher et al., 2013), but they were not designed to prove a WSG-mediated mechanism.

Unexpected is the insignificant effect of LAI on WP in both transects, visible in the exclusion of leaf area from the models. This is difficult to reconcile with the fact that canopy carbon gain depends on the amount of intercepted light and that the LAI-productivity relationship is usually tight in forests (Reich 2009). A possible explanation could be that LAI measurements with

the LAI2000 Canopy Analyzer are known to lead to both under-estimation where leaf clumping is dominant, and over-estimation where stem density is high. This may have blurred LAI differences between stands differing in productivity.

Tree diversity apparently has only a small influence on productivity in the studied tropical montane forests, even though tree species numbers varied largely across the plots (6-27 species per 0.04 ha) and plot size was small, so that the effect should be better visible. A positive diversity effect apparently was confined to the component fine litter production (in the Loja transect), as the effect became more visible when $NPP_a$ (which includes fine litter production) was considered and not wood production alone (Figure 4). A possible explanation of the weak diversity effect is that the species richness in our plots may have been too high to show a positive DPR, as diversity effects on productivity generally are more pronounced at lower species numbers (Paquette & Messier, 2011; Tilman et al., 2012). It would be interesting to see how tree species richness translates into functional diversity in our two transects and if functional diversity is already saturating at intermediate levels of tree diversity (as a result of increasing functional redundancy). That would explain the weak diversity effect. But functional trait data is only available for a small fraction of the recorded tree species.

Our results support our third hypothesis and fit the controversial picture that emerges from the existing observational studies on the DPR in tropical forests. Chisholm et al. (2013) systematically explored the role of diversity for productivity in natural tropical forests and found only a small absolute effect: When controlling for stem density effects in the 0.04 ha plots, a doubling of species richness corresponded in their study only to a 5% increase in WP on average. At larger scale (0.25 ha, 1 ha), results were mixed and negative relationships became more common. This points at a negligible role of tree diversity for productivity in tropical forest landscapes, in line with our findings. These results from tropical forests match findings from more than 100,000 forest plots in the United States, where the predominant DPR was concave-negative in humid climates and primarily non-significant in harsh climates (Fei et al., 2018). Similar to the DPR, a significant positive diversity–aboveground biomass relationship was detectable in tropical lowland forests only in small plots (0.04 ha), while it disappeared at larger scale (1 ha plots) (Sullivan et al., 2017). Moreover this study showed that the diversity effect on AGB was small: Doubling species richness at 0.04 ha increased biomass only by 6.9%.

An additional important result of our study is the high variation in stand properties and WP across the plots of an elevation level (Figure 2). Environmental variation related to topography can be a strong driver of forest structure and tree species composition, and it influences productivity and nutrient cycling through changes in traits, as has been shown for several tropical lowland forests (Fortunel, et al. 2018, Jucker et al. 2018), and also for the TMF of the Loja transect (Werner & Homeier, 2015; Pierick et al. 2020).

Our analysis of multiple drivers of productivity further shows that a diversity effect, even if significant, is only of minor importance, when other productivity-influencing factors are also included in the analysis. In our case, diversity ranged behind effects of elevation (temperature), soil nutrient availability and wood specific gravity. For understanding the controls of forest productivity, it is more important to study the influences of environmental factors and stand structural and functional properties in more detail.

## 5 Conclusions

From the analysis of more than 80 old-growth forest plots in two highly diverse Andean regions with large geological and topographic heterogeneity, we conclude that the main determinants of aboveground productivity in tropical montane forests are elevation (primarily as a proxy of temperature), nutrient supply and stand structural properties, given non-limiting moisture conditions. Nutrient availability seems to affect productivity not only directly but also indirectly through stand properties, while tree diversity has only a small influence on productivity. Tree species turnover and associated changes in functional traits are far more important for productivity than changes in species diversity. Functional biodiversity research has to proceed from the search for the significance of diversity effects, which has dominated biodiversity–ecosystem functioning (BEF) research in the past, to an assessment of the importance of diversity effects for productivity and other ecosystem processes in order to better understand the functional role of diversity. A deeper understanding of foliar and root trait variation with elevation and among the species of a community will help to achieve a more mechanistic understanding of tropical forest productivity change along environmental gradients. Our findings are of practical relevance, as the analysis addresses spatial scales relevant for old-growth forest conservation and management.

## Appendix A: Study sites and plots

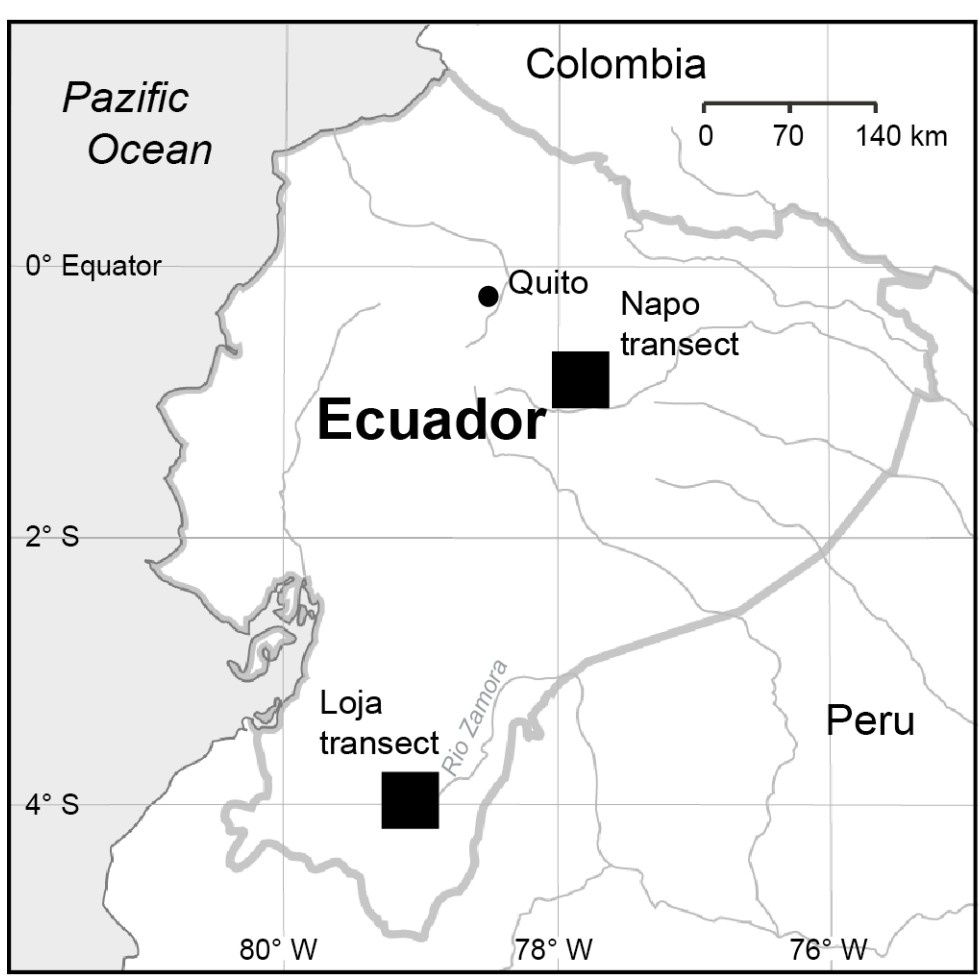

**Figure A1.** Map showing the location of the two study sites in Ecuador.



## Appendix B: PCAs of soil properties

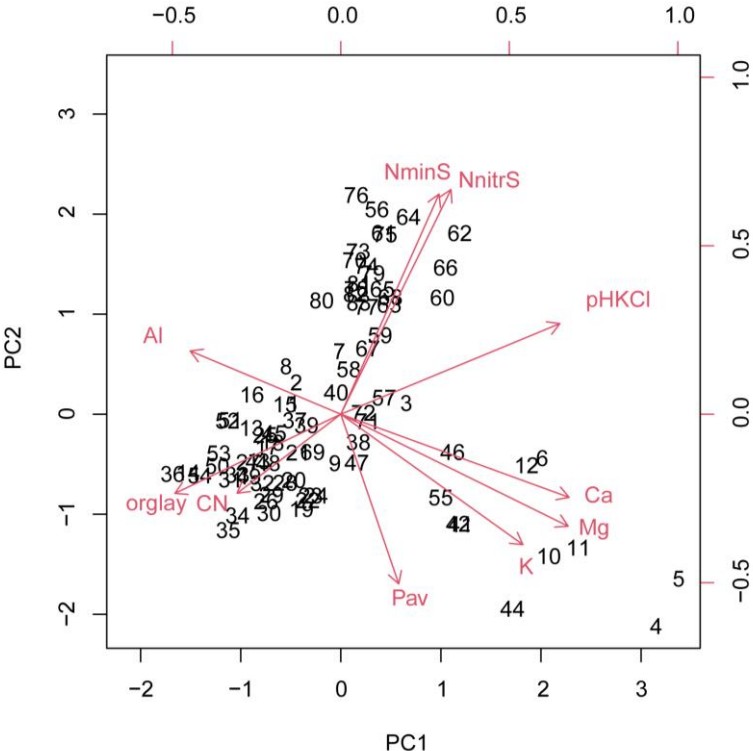

**Figure B1.** Principal component analysis of the soil properties of 83 study plots (Loja transect and Napo transect), PC1 and PC2 explained 37.4% and 26.1% of variance, respectively.

**Table B1.** Factor loadings of the variables in the PCA (Figure B1).

| Variable | PC1 | PC2 |
|---|---|---|
| org. layer depth | -0.319 | -0.182 |
| pH KCl | 0.421 | 0.208 |
| K | 0.349 | -0.300 |
| Mg | 0.436 | -0.258 |
| Ca | 0.438 | -0.192 |
| Al | -0.289 | 0.146 |
| CN | -0.199 | -0.181 |
| Pav | 0.111 | -0.390 |
| NminS | 0.189 | 0.507 |
| NnitrS | 0.212 | 0.517 |

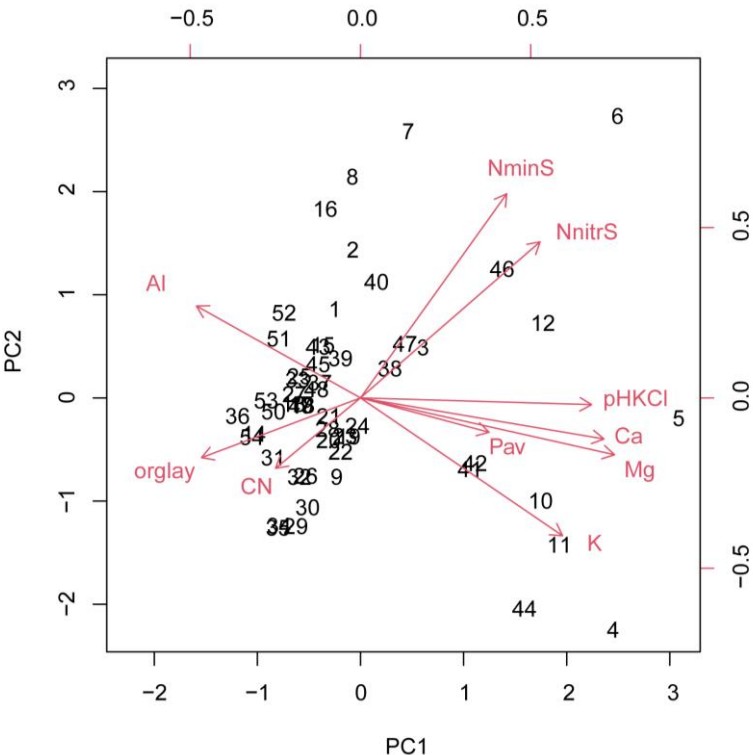

**Figure B2.** Principal component analysis of the soil properties of 54 study plots of the Loja transect, PC1 and PC2 explained 46.9% and 14.6% of variance, respectively.

**Table B2.** Factor loadings of the variables in the PCA (Figure B2).

| Variable | PC1 | PC2 |
|---|---|---|
| org. layer depth | -0.270 | -0.182 |
| pH KCl | 0.392 | -0.020 |
| K | 0.343 | -0.419 |
| Mg | 0.431 | -0.173 |
| Ca | 0.413 | -0.125 |
| Al | -0.278 | 0.280 |
| CN | -0.144 | -0.214 |
| Pav | 0.218 | -0.104 |
| NminS | 0.248 | 0.621 |
| NnitrS | 0.305 | 0.475 |


**Data availability**

All relevant data are within the paper and its appendices.

**Supplement**

Supplement S1: Plot forest parameters

Supplement S2: Plot soil parameters

**Author contribution**

JH and CL designed the study, JH performed research and analyzed the data, JH and CL wrote the manuscript.

**Competing interests**

The authors declare that they have no conflict of interest.

**Acknowledgements**

We thank the universities in Loja (UTPL, UNL) and in Quito (PUCE) for continuous support during our field studies. We thank the Ministerio de Ambiente del Ecuador for granting the research permits. We further acknowledge the assistance of Nixon Cumbicus, Miguel-Angel Chinchero, Jaime Peña, Roman Link, Malte Unger and Katrin Mikolajewski during field and lab work. The funding provided by the Federal Ministry of Education and Science (BMBF) within the Pro Benefit project and

by the German Research Foundation (DFG) through research grants Ho3296/2, Ho3296/4 and Le762/10 is gratefully acknowledged. We thank two anonymous reviewers for their constructive criticism of an earlier draft of the manuscript.

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
