# Peer review of "Factors controlling the productivity of tropical Andean forests: Climate and soil are more important than tree diversity"

_Biogeosciences, 2020_

## Referee Comment (RC1) · Anonymous Referee #1 · 7 Oct 2020

General

This manuscript presents an analysis of various factors on wood productivity and net primary productivity across a series of plots located on two transects. Although the findings appear robust, logically, and technically correct, I believe the analysis could be improved by better describing key details like the calculation of wood productivity, inclusion of additional covariates (particularly stand structural attributes), and general model behavior as well as fit statistics. In addition, a few paragraphs in the Introduction could be further expanded with key details.

Specific

[Figure]

L14: How is "productivity" being defined here? ANPP?

L50-53: Seems this paragraph and a few the other ones above it should be further expanded? How widespread are tropical montane forests? Where are they primarily located? Why specifically focus on them?

L57: Don't understand the use of "rarefied" here.

L60: I am confused by the "10 K" Can this be presented differently?

L67: TMF was not previously defined and I assume referring to tropical montane forests?

L106: Some additional details would be helpful here. I assume these are predicted biomass values? What was the average remeasurement length? Is annual AGB increment computed from tree rings?

L130-133: I am bit confused by this. Personally, I would use AGB to predict WP or NPP, while I would consider WSG to be more of a function of species composition than stand structure? Seems other structural attributes could be computed like total basal area, quadratic mean diameter, and measures of the diameter distribution?

L141: What are RMSEA and CFI?

Figure 2: Might not include 0 on graphs with narrow distributions like LAI and WSG to better highlight trends.

L276: Your LAI cover a very narrow range and often the strong relationships are observed when values are below 5-6.

---

## Referee Comment (RC2) · Anonymous Referee #2 · 8 Dec 2020

Review of Homeier and Leuschner Biogeosciences.

Factors controlling the productivity of tropical Andean forests: Climate and soil are more important than tree diversity

This paper is a very exciting summary of a large number of 400 m2 forest plots distributed over a large elevation gradient in the Andes. It addresses questions relating to productivity, species diversity and environment. It is rare for a dataset of this size to be assembled, and that alone justifies support for publication, after revision. The focus of the paper starts with a discussion of the diversity-function debate and then moves to a discussion on the determinants of low productivity at high elevation in tropical montane forests. Both subjects are dealt with usefully but I think both need some attention. The theoretical importance of the diversity-function debate is mentioned, but if this line of argument is to be maintained, I think it needs further justification. The debate has practical importance, but the theoretical importance, whilst it exists, receives a fair bit of skepticism in the literature. I don't think this point is central to the main strength of the paper, but I mention it, as the authors may choose to soften the stance on the theoretical importance or to support it more fully. Further on this point, I note that some large observational datasets are referred to, in order to address diversity-function relationships across biome types. I am supportive of this; all approaches to the diversity-function question have some unavoidable flaws in relation to this question but the experimental ones seem most prone to it. On the other hand some key experimental work in the tropics is not mentioned – surely it is of interest to place this discussion in the context of the Sabah Biodiversity Experiment – a large scale experiment in lowland tropical forest? There are few such comparisons in tropical forest to make use of and this feels like a gap.

On the question of causes/predictors of reduced productivity at high elevation, the comparative and interpretive analysis seems a little limited, even though the data are impressive and wide-ranging. The authors note the work of Fyllas 2017 (Ecology Letters), where a modelling approach was taken using annual estimates of GPP along a Peruvian (ie also in the Andes) elevation gradient as validation data. However, Homeier et al miss the first mechanistic modelling study of tropical montane forest productivity differences by elevation, presented by van der Weg et al. 2014 (Ecosystems). The 2014 paper validates model estimates of productivity using fine-scale mechanistically-related detailed sap flux data, whilst the 2017 paper validates mainly against impressive annual GPP estimates based on summed measurements of NPP and respiration. They come to different conclusions. Based on Amax values measured above 20 deg C, Fyllas et al highlight the importance of variation in Amax with elevation and variation in radiation with elevation, whilst van der Weg et al using Vcmax/Jmax, a stomatal model and measured leaf temperature (shown to be frequently well below 20 C in the high elevation site), conclude that variations in temperature and radiation are the most important drivers. The Fyllas conclusion is attractive as it suggests that despite high species turnover, overall Amax 'responds' by increasing at lower temperatures, suggesting a degree of 'optimisation to environment', filtered by species turnover.  On the other hand, the mechanistic validation in the van der Weg paper, and its use of real leaf temperatures well below 20 C suggests a key role for temperature not strongly evident in the Fyllas analysis. The paper here (Homeier et al) could contribute strongly to this overall discussion with independent data and analysis. Whilst the density of measurements is not the same as presented in Fyllas, and there is no modelling (which is not necessarily a problem), there is very detailed edaphic information, as well as productivity and information on species identity. It is also clear that there is a huge range in productivity at each elevation among the different forest plots. This may be noise related (smallish plots…but lots of them!) or it may be environmental; I note that the data Fyllas et al use for Amax values suggest a wide range of photosynthetic capacities at each elevation….ie a similar pattern as found here, of

much variation at each elevation. In sum, this paper has the potential to make a bigger contribution to this discussion than it currently does. I hope these comments are of use; it is worth expending effort on in a revision because the question is about fundamental tropical ecology (or indeed montane-to-lowland ecology), and had remained in the realm of 'many explanatory factors but we don't know which' until relatively recently.

Detailed points

Line 28. What about reference to the Sabah Biodiv Expt? Can you make more of the comparison of the diversity-function relationship at high vs low diversity? Also, is it useful to refer to Sullivan 2016 Scientific Reps (biomass and diversity…very slightly different question as biomass is not 'function' but it does discuss plot size)

Line 37-45. Missing the van der Weg 2014, which used data and modelling in early trop montane forest data+modelling analysis. It shows temp and radiation dependence mechanistically, and water use is validated using sap flux data. It also demonstrates importance of low leaf temps affecting function. The analysis in these lines mentions some key comparative flux and parameter data (eg Girardin 2014 Malhi 2017), but omits the Fyllas 2017 paper mentioned above. A bigger discussion is needed somewhere here to set this paper up more comprehensively.

Line 47. I note the use of the Chisholm 2013 reference, but I think this analysis/discussion needs to take into account dynamics too, if only briefly. ABG does always reflect productivity - see Baker 2004, Galbraith et al. 2013, Malhi et al. 2015); residence time is important, as is recruitment. So the point made here needs to be made in relation to this wider discussion on determinants of ABG.

Line 59. The choice and effectiveness of small plots needs a fuller discussion than reference only to Chisholm. For example, if diversity effects are only likely to emerge at small scale, what does this mean for their fundamental role?

Line 135. It's great to see a careful path analysis approach being taken here to distinguish different drivers. However, can you explain how the original model structure affected the ultimate outcomes? Might you have had a different outcome had your starting point (structure) been different or have your methods fully accounted for this? (apologies if I've missed this point).

Line 160. Are there any recruitment data to advance the C dynamics analysis?

Line 164. LAI measurements are really important but difficult to make. Are you sure the differences you see in LAI are related to leaf area and not a change in stem density/canopy structure? High stem density would increase Plant Area Index (ie leaves and wood) even if LAI did not increase. I know this is hard to separate, but some comment/discussion/caveat would be useful.

Fig 2. There are strong signals of variation in mean values with elevation in some of the key metrics (eg WP, Stem density). But there is also very large variation at each elevation. is this discussed? The variation by elevation is larger than the overall mean signal in the regression; this has also been observed in ecophys measurements elsewhere (eg Bahar et al. 2016 New Phytologist).

Line 193. As per the second paragraph above, the Fyllas 2017 paper notes that there is large turnover in species but argues that there are directional changes in mean trait values with elevation and these become determining of productivity along with radiation....how can we link these different findings?

Line 237. The role of low leaf temperatures needs further consideration in affecting rates of carbon gain, not just radiation levels.

Line 243. This soils dataset is very substantive and provides detailed driver information for the path analysis. It may be possible to use this to help the contrast with or discussion of preceding analyses on this general productivity/elevation subject.

Line 250. Do you have soil respiration data or root productivity data to back this up (ie higher allocation of C to root production/symbionts)?

Line 265. I wonder if an analysis of productivity vs biomass would help here too – ie productivity does not determine biomass in all circumstances because of other fluxes/processes affecting C residence time.

Line 268, it seems natural to consider a comparison with the effects of fertilisation in the Andes reported by Fisher et al. 2013, Oecologia, as well as this 1989 reference.

Line 274 – as before, please consider LAI vs PAI differences, causative factors.

Line 280. I wonder whether this section could be given a bit more depth by including a discussion on the relationship between trait diversity and species diversity? Might we expect a stronger response at low species diversity because trait diversity may increase rapidly as you add species at first, but if trait diversity is ultimately lower than species diversity we might expect the function-diversity graph to saturate more quickly using traits? Also of course there is the wider discussion on how the relationship (with species) varies under harsh and less harsh environments (eg. Paquette et al. 2011, Glob Ecol and Biogeog).

Line 291. Again, might some discussion on traits be useful here too?

---

## Author Comment (AC1) · 21 Dec 2020

Response to Reviewer #1

Reviewer#1: This manuscript presents an analysis of various factors on wood productivity and net primary productivity across a series of plots located on two transects. Although the findings appear robust, logically, and technically correct, I believe the analysis could be improved by better describing key details like the calculation of wood productivity, inclusion of additional covariates (particularly stand structural attributes), and general model behavior as well as fit statistics. In addition, a few paragraphs in the Introduction could be further expanded with key details.

Answer: Thank you for the helpful comments to our manuscript! We have now expanded the Methods section on the measurement of biomass and productivity and include more details on our calculations. The concept, where the SEM is based on, included elevation, tree diversity, soil and stand properties as predictors of productivity. We included only AGB and WSG (WSG showed a stronger correlation to stand productivity than LAI or stem density) in the SEM. Additional structural variables such as basal area or quadratic mean diameter in the model would have weakened the analysis, as they are closely related to AGB. Model fit statistics are given in the figure legends (Figure 4 and 5). The discussion of possible abiotic and biotic drivers of forest productivity in the Introduction has been expanded, as recommended.

Reviewer#1: L14: How is "productivity" being defined here? ANPP?

Answer: We changed it to "wood production" (result of the overall analysis of all plots), because ANPP was only analyzed for the Loja transect.

Reviewer#1: L50-53: Seems this paragraph and a few the other ones above it should be further expanded? How widespread are tropical montane forests? Where are they primarily located? Why specifically focus on them?

Answer: We are now introducing tropical montane forests as an ecosystem type in more detail.

Reviewer#1: L57: Don't understand the use of "rarefied" here.

Answer: We replaced "rarefied number of tree species per plot" by "tree diversity", the rarefaction method is explained in detail in the data analysis paragraph.

Reviewer#1: L60: I am confused by the "10 K" Can this be presented differently?

Answer: K (degrees Kelvin) is the SI unit for temperature differences; it should be used instead of °C, when differences are meant.

Reviewer#1: L67: TMF was not previously defined and I assume referring to tropical

montane forests?

Answer: We define TMF now earlier in the Introduction.

Reviewer#1: L106: Some additional details would be helpful here. I assume these are predicted biomass values? What was the average remeasurement length? Is annual AGB increment computed from tree rings?

Answer: The plot biomass values were calculated for each plot as the sum of the biomass of the single stems using the Chave et al (2005) equation for tropical wet forests with stem diameter, wood specific gravity (WSG) and tree height as parameters. Re-measurement intervals were between 1 and 5 years, depending on the study sites. We describe the biomass and wood production measurements now in some more detail.

Reviewer#1: L130-133: I am bit confused by this. Personally, I would use AGB to predict WP or NPP, while I would consider WSG to be more of a function of species composition than stand structure? Seems other structural attributes could be computed like total basal area, quadratic mean diameter, and measures of the diameter distribution?

Answer: We also used AGB as a predictor for WP (see Figure 4), in addition, we selected WSG from the stand properties (LAI, stem density, WSG) because it showed a stronger correlation to stand productivity than LAI and stem density. Both basal area and quadratic mean diameter are highly correlated to AGB, and we think that AGB is the most meaningful of these variables. We changed "stand structural variables" to "stand properties" in the respective sentence.

Reviewer#1: L141: What are RMSEA and CFI?

Answer: RMSEA (root mean square error of approximation) and CFI (comparative fit index) were used to assess the goodness of model fit.

Reviewer#1: Figure 2: Might not include 0 on graphs with narrow distributions like LAI

and WSG to better highlight trends.

Answer: The respective figures are improved to make the elevational trend more visible.

Reviewer#1: L276: Your LAI cover a very narrow range and often the strong relationships are observed when values are below 5-6.

Answer: We now discuss at the end of the Discussion the assumed shortcomings of optical LAI estimates in complex forests and refer to stems and branches, which are recorded by the LAI2000 systems as well. Litter trapping studies in several plots in the Loja transect confirm these assumptions about under- and overestimation of LAI by optical methods.

References: Chave J et al. (2005) Tree allometry and improved estimation of carbon stocks and balance in tropical forests. Oecologia 145:87–99

---

## Author Comment (AC2) · 23 Dec 2020

Response to Reviewer #2

Reviewer#2: This paper is a very exciting summary of a large number of 400 m2 forest plots distributed over a large elevation gradient in the Andes. It addresses questions relating to productivity, species diversity and environment. It is rare for a dataset of this size to be assembled, and that alone justifies support for publication, after revision. The focus of the paper starts with a discussion of the diversity-function debate and then moves to a discussion on the determinants of low productivity at high elevation in tropical montane forests. Both subjects are dealt with usefully but I think both need some

attention. The theoretical importance of the diversity-function debate is mentioned, but if this line of argument is to be maintained, I think it needs further justification. The debate has practical importance, but the theoretical importance, whilst it exists, receives a fair bit of skepticism in the literature. I don't think this point is central to the main strength of the paper, but I mention it, as the authors may choose to soften the stance on the theoretical importance or to support it more fully. Further on this point, I note that some large observational datasets are referred to, in order to address diversity-function relationships across biome types. I am supportive of this; all approaches to the diversity-function question have some unavoidable flaws in relation to this question but the experimental ones seem most prone to it. On the other hand some key experimental work in the tropics is not mentioned – surely it is of interest to place this discussion in the context of the Sabah Biodiversity Experiment – a large scale experiment in lowland tropical forest? There are few such comparisons in tropical forest to make use of and this feels like a gap.

Answer: Thank you for the detailed suggestions to our manuscript and for this thoughtful comment on the biodiversity-function relationship. We have rewritten part of the Introduction with relation to DPR and have softened the wording in places. We now mention shortcomings of both experimental approaches and observational studies on the DPR. In addition to the Sardinilla experiment (Panama; Schnabel et al. 2019), we now also mention the Sabah Biodiversity Experiment (Tuck et al. 2016) – however, due to the young age of the trees, no analysis of overyielding has apparently been published so far.

Reviewer#2: On the question of causes/predictors of reduced productivity at high elevation, the comparative and interpretive analysis seems a little limited, even though the data are impressive and wide-ranging. The authors note the work of Fyllas 2017 (Ecology Letters), where a modelling approach was taken using annual estimates of GPP along a Peruvian (ie also in the Andes) elevation gradient as validation data. However, Homeier et al miss the first mechanistic modelling study of tropical montane forest productivity differences by elevation, presented by van der Weg et al. 2014 (Ecosystems). The 2014 paper validates model estimates of productivity using fine-scale mechanistically-related detailed sap flux data, whilst the 2017 paper validates mainly against impressive annual GPP estimates based on summed measurements of NPP and respiration. They come to different conclusions. Based on Amax values measured above 20 deg C, Fyllas et al highlight the importance of variation in Amax with elevation and variation in radiation with elevation, whilst van der Weg et al using Vcmax/Jmax, a stomatal model and measured leaf temperature (shown to be frequently well below 20 C in the high elevation site), conclude that variations in temperature and radiation are the most important drivers. The Fyllas conclusion is attractive as it suggests that despite high species turnover, overall Amax 'responds' by increasing at lower temperatures, suggesting a degree of 'optimisation to environment', filtered by species turnover. On the other hand, the mechanistic validation in the van der Weg paper, and its use of real leaf temperatures well below 20 C suggests a key role for temperature not strongly evident in the Fyllas analysis. The paper here (Homeier et al) could contribute strongly to this overall discussion with independent data and analysis. Whilst the density of measurements is not the same as presented in Fyllas, and there is no modelling (which is not necessarily a problem), there is very detailed edaphic information, as well as productivity and information on species identity. It is also clear that there is a huge range in productivity at each elevation among the different forest plots. This may be noise related (smallish plots. . .but lots of them!) or it may be environmental; I note that the data Fyllas et al use for Amax values suggest a wide range of photosynthetic capacities at each elevation. . ..ie a similar pattern as found here, of much variation at each elevation. In sum, this paper has the potential to make a bigger contribution to this discussion than it currently does. I hope these comments are of use; it is worth expending effort on in a revision because the question is about fundamental tropical ecology (or indeed montane-to-lowland ecology), and had remained in the realm of 'many explanatory factors but we don't know which' until relatively recently.

Answer: Thank you for the detailed suggestions. We have extended the discussion

about temperature effects on productivity in TMF by including various additional papers, notably van de Weg et al. 2014 and the modeling study of Fyllas et al. 2017, and also the empirical study of elevational change in photosynthesis of Wittich et al. 2012. In the Discussion section, possible pathways through which temperature could influence tree metabolism are now mentioned and part of the Discussion has been rewritten. As a possible indirect low-temperature effect on physiology, the slowing down of N mineralization and resulting nutrient deficiency is discussed. The possibility that temperature manifests primarily through trait variation due to the elevational species turnover is also mentioned. We added two sentences about the importance of environmental variation within elevation levels to the last paragraph of the discussion.

Detailed points Reviewer#2: Line 28. What about reference to the Sabah Biodiv Expt? Can you make more of the comparison of the diversity-function relationship at high vs low diversity? Also, is it useful to refer to Sullivan 2016 Scientific Reps (biomass and diversity. . .very slightly different question as biomass is not 'function' but it does discuss plot size)

Answer: The Sabah Biodiv Experiment is mentioned now (Tuck et al. 2016), even though we are not aware of a study reporting productivity data that prove overyielding in this experiment.

Reviewer#2: Line 37-45. Missing the van der Weg 2014, which used data and modelling in early trop montane forest data+modelling analysis. It shows temp and radiation dependence mechanistically, and water use is validated using sap flux data. It also demonstrates importance of low leaf temps affecting function. The analysis in these lines mentions some key comparative flux and parameter data (eg Girardin 2014 Malhi 2017), but omits the Fyllas 2017 paper mentioned above. A bigger discussion is needed somewhere here to set this paper up more comprehensively.

Answer: The van de Weg et al. 2014 paper is now mentioned in the Discussion, as is the Fyllas et al. paper. The overview of possible abiotic and biotic controls of TMF

productivity has been expanded in the Introduction.

Reviewer#2: Line 47. I note the use of the Chisholm 2013 reference, but I think this analysis/discussion needs to take into account dynamics too, if only briefly. ABG does always reflect productivity - see Baker 2004, Galbraith et al. 2013, Malhi et al. 2015); residence time is important, as is recruitment. So the point made here needs to be made in relation to this wider discussion on determinants of ABG.

Answer: The relation between ABG and ANPP and the role of woody tissue residence time are now dealt with in more detail in the Introduction. It is mentioned that the relation is often weak (even though some studies reported an AGB effect on wood production).

Reviewer#2: Line 59. The choice and effectiveness of small plots needs a fuller discussion than reference only to Chisholm. For example, if diversity effects are only likely to emerge at small scale, what does this mean for their fundamental role?

Answer: We now discuss the advantages of small plots in rugged terrain, with respect to the chance of detecting diversity effects and concerning time consumption and the resulting potential to monitor a larger number of plots in total. At the end of the Discussion, we mention that diversity effects that manifest in 0.04 ha plots may be of low relevance for the landscape level.

Reviewer#2: Line 135. It's great to see a careful path analysis approach being taken here to distinguish different drivers. However, can you explain how the original model structure affected the ultimate outcomes? Might you have had a different outcome had your starting point (structure) been different or have your methods fully accounted for this? (apologies if I've missed this point).

Answer: The reviewer is right, SEM model structure affects the outcome. Therefore we carefully developed a structure that, based on the available data and our knowledge, represented the best combination of predictors for productivity (elevation, tree diversity,

soil and stand properties). We iteratively removed insignificant paths from the starting model to test whether incorporation of those paths improved the model fit (described in the last paragraph of the data analysis chapter).

Reviewer#2: Line 160. Are there any recruitment data to advance the C dynamics analysis?

Answer: For our study we focused on stem diameter growth of the surviving trees, assuming an equivalence of stem mortality and recruitment. We think that to determine reasonable rates of mortality and recruitment would require bigger plots.

Reviewer#2: Line 164. LAI measurements are really important but difficult to make. Are you sure the differences you see in LAI are related to leaf area and not a change in stem density/canopy structure? High stem density would increase Plant Area Index (ie leaves and wood) even if LAI did not increase. I know this is hard to separate, but some comment/discussion/caveat would be useful.

Answer: We now discuss at the end of the Discussion the assumed shortcomings of optical LAI estimates in complex forests and refer to stems and branches, which are recorded by the LAI2000 system as well. Litter trapping studies in several plots in the Loja transect confirm these assumptions about under- and overestimation of LAI by optical methods.

Reviewer#2: Fig 2. There are strong signals of variation in mean values with elevation in some of the key metrics (eg WP, Stem density). But there is also very large variation at each elevation. Is this discussed? The variation by elevation is larger than the overall mean signal in the regression; this has also been observed in ecophys measurements elsewhere (eg Bahar et al. 2016 New Phytologist).

Answer: We fully agree. This is an important topic, and we have addressed variation in fine root traits at a given elevation in the Loja transect in a separate paper that went online in New Phytologist recently (Pierick et al. 2020). We added two sentences about

the importance of environmental variation within elevation levels to the last paragraph of the discussion.

Reviewer#2: Line 193. As per the second paragraph above, the Fyllas 2017 paper notes that there is large turnover in species but argues that there are directional changes in mean trait values with elevation and these become determining of productivity along with radiation....how can we link these different findings?

Answer: Thank you, this fits perfectly with what we found for leaf traits in the manuscript by Homeier et al. (to be submitted). We mention now the assumption that trait variation due to elevational species turnover explains a large part of the temperature effect on productivity (and photosynthesis). We mention the Fyllas et al. paper here as well.

Reviewer#2: Line 237. The role of low leaf temperatures needs further consideration in affecting rates of carbon gain, not just radiation levels.

Answer: We agree. See response above. The somewhat contradictory evidence with respect to direct and indirect temperature effects on tree metabolism in Andean TMF is discussed in more detail now in both the Introduction and the Discussion.

Reviewer#2: Line 243. This soils dataset is very substantive and provides detailed driver information for the path analysis. It may be possible to use this to help the contrast with or discussion of preceding analyses on this general productivity/elevation subject.

Answer: Our finding that not only N and P, but likely also basic cations are influencing productivity in an independent manner, is now discussed with respect to the widespread assumption of P limitation in lowland forests and N limitation in high-elevation forests.

Reviewer#2: Line 250. Do you have soil respiration data or root productivity data to back this up (ie higher allocation of C to root production/symbionts)?

Answer: We do not have such data for the complete plot set investigated here. But

earlier publications on fine root biomass, fine root production (using both minirhizotron observations and sequential coring approaches) and elevational allocation shifts in the Loja transect help to interpret the findings presented here. We mention now these publications.

Reviewer#2: Line 265. I wonder if an analysis of productivity vs biomass would help here too – ie productivity does not determine biomass in all circumstances because of other fluxes/processes affecting C residence time.

Answer: Since we were primarily interested in productivity as a dependent variable, we did not study effects of productivity on AGB (which are of course existing) but rather AGB effects on productivity. We feel that studying the NPP-AGB relation in more detail would inflate the manuscript; it should better be performed in a separate analysis together with mortality data.

Reviewer#2: Line 268, it seems natural to consider a comparison with the effects of fertlisation in the Andes reported by Fisher et al. 2013, Oecologia, as well as this 1989 reference. Answer: The fertilization experiments in Andean forests of Peru and Ecuador are mentioned now.

Reviewer#2: Line 274 – as before, please consider LAI vs PAI differences, causative factors. Answer: See above. We discuss the potential LAI measurement errors at the end of the Discussion.

Reviewer#2: Line 280. I wonder whether this section could be given a bit more depth by including a discussion on the relationship between trait diversity and species diversity? Might we expect a stronger response at low species diversity because trait diversity may increase rapidly as you add species at first, but if trait diversity is ultimately lower than species diversity we might expect the function-diversity graph to saturate more quickly using traits? Also of course there is the wider discussion on how the relationship (with species) varies under harsh and less harsh environments (eg. Paquette et al. 2011, Glob Ecol and Biogeog).

Answer: It would be interesting to see how tree species richness translates into functional diversity in our transects. Unfortunately, functional trait data is so far only available for a small fraction of the tree species from both transects. We added three sentences, offering saturating functional diversity and increasing functional redundancy as a possible explanation for the weak diversity effect.

Reviewer#2: Line 291. Again, might some discussion on traits be useful here too?

Answer: See above.

---

## Author Response (AR2)

**Response to Associate Editor**

**Associate Editor Decision: Publish subject to minor revisions (review by editor)** (08 Jan 2021) by [Sara Vicca]
Comments to the Author:

Dear authors,

Your revised manuscript has now reviewed by the reviewer, who is very positive about the dataset that you present but pointed out that the manuscript could still be considerably improved by further elucidating the mechanisms behind the observations. I agree with this and think the reviewer provided very useful input for you to further improve the manuscript.

I would also like to add my appreciation about the dataset and especially for including and providing multiple soil data. I did miss some motivation for the choice of soil measurements - why these and not other soil properties and nutrient measurements. Please include that information in the revised manuscript and also indicate (perhaps in the discussion) if there are any potentially important variables that may still be missing from the dataset. From my experience, soil organic matter content is an important variable that is not included in your dataset. Organic layer depth is probably a good proxy and this could be mentioned explicitly in the text. Soil texture is another important soil property that is missing from the dataset. Do you have any indication if texture varied (substantially) across the gradient?

Please consider all suggestions carefully and in case you choose not to implement some of them, provide arguments for that in the response letter.

I look forward to receiving your revised manuscript.

Kind regards,
Sara Vicca

**Answer:** Thank you for the constructive criticism!

We think, that we already included more soil parameter than most other tropical forest transect studies, But you are right, there are some additional parameters missing that probably might improve our understanding of soil effects on forest productivity. Unfortunately, data on soil texture and soil organic matter content are not available (or only for a subset of plots). We assumed that soil hydrology in this perhumid climate is of secondary importance for tree growth and therefore soil texture might be less important than the studied soil parameters.

We explain in the Methods (and also the Discussion) that we selected the physiologically most meaningful soil chemical properties. These are the five plant macronutrients (N, P, K, Ca, Mg) in their plant-available form (N supply rate, plant-available P, salt-exchangeable Ca, K, Mg), in addition C/N ratio as an indicator of decomposability, and soil acidity.

The following sentence was added to the Methods section: The increase in organic layer depth with elevation in both transects, accompanied by wider C/N ratios and lower pH, is

probably related to higher soil organic matter contents and indicates a decrease in nutrient availability as a result of reduced organic matter turnover.

**Response to Reviewer #2**

**Reviewer#2:** The revision and response letter addresses all the points raised; thanks very much for the effort, and also for the analysis of an important and fascinating dataset. Overall the ms is improved, but there are some key issues that have not really been elucidated enough here for publication, leaving the analysis much less complete than it should or could be. There is the odd useful but missing reference that I think would be helpful to add, but the main conceptual gap appears to be around the connection of temperature with mechanism and the relationships to traits and diversity.

The authors mention a plot size effect on analyzing diversity/productivity relationships. I thought I had mentioned the paper of Sullivan 2016 (Sci Reps) before, but perhaps not (apologies if not). This is relevant to the introductory material around line 28 and in the discussion, around line 345-355: Sullivan et al. 2016 find (small) plot size can influence inferences from their analysis of diversity and AGB/C storage in tropical forests. This needs to be included/discussed, I think.

**Answer:** We initially refrained from citing the Sullivan et al. (2016) study here, because it addresses the diversity – biomass relationship (DBR) and not the diversity – productivity relationship (DPR), which is our focus. The relation between aboveground biomass and productivity is weak in tropical forests, as carbon residence time is a key determinant of standing biomass, which varies considerably with climate and soil. We now mention Sullivan et al. (2016) in the Discussion related to plot size effects on the DPR.

**Reviewer#2:** As for the productivity-temperature-elevation discussion this seems to (i) miss some of the earlierst references to the subject (eg S Bruijnzeel's classic work) and (ii) be a bit simplified to the point that key issues are obscured (or inadvertently lost) in the current text. There is significant advantage in being specific here – eg TMF or tropical lowland forest studies often focus on NPP (or just tree growth, as here), but others also consider GPP, both from an empirical and modelling perspective, and the papers in the introduction (eg Line 44-60) refer to several different versions of 'productivity'.

**Answer:** We added a key publication of Bruijnzeel & Veneklaas (1998). Moreover, we rewrote part of the Introduction section, which focuses on assumed temperature effects on productivity. We stress that temperature may have direct and indirect effects on productivity. All cited work refers to NPP, and not GPP, now.

**Reviewer#2:** One advantage of being specific is that it allows the author to dissect better the drivers of reduced productivity (at higher elevation), especially when linking tree growth and gross primary productivity (=gross photosynthesis, GPP). This is can be powerfully done using a combination of data and mechanistic modelling. Line 46 introduces modelling work but refers only to that of Fyllas 2017. The Fyllas study uses a mechanistic model, but validates only based on annual-scale estimates of summed components of the carbon cycle. The modelling study of van der Weg 2014 is not mentioned except in reference to LMA, and yet this earlier modelling paper (the first to use a mechanistic analysis to identify

temperature as the main driver of differences in GPP with elevation in TMF) validates model results based on high-frequency sap flux data as well as annual-scale component-summed carbon cycle analysis. That is, it provides validation data at a timescale relevant to the processes being driven by variation in the environment, and via the relevant mechanisms. Both the Fyllas and van der Weg papers use photosynthetic physiology at the core of their models but they come up with different interpretations of the main driver of change productivity (GPP) with elevation. As indicated in the first review, this needs to be discussed because the van der Weg findings are consistent with those in this submitted paper (ie that temperature is a dominant driver) but the Fyllas paper suggests that any effects of temperature are expressed through trait variation.

Taking the analysis presented in this submitted manuscript on Ecuadorian data a step further, the work of Wittich 2012 is cited as showing that light-saturated assimilation rates (called Amax) are lower at higher elevation, ie this is consistent with the results presented in this submitted paper (of lower wood production (WP)) at higher elevation...but I don't know if the Wittich values are temperature-corrected, or if they are cited at ambient temperature (ie T differing by elevation).

**Answer:** The Amax measurements of Wittich et al. (2012) were taken at ambient temperature of the respective elevation (typical temperature at noon on a sunny day); thus they reflect the elevational temperature decrease. The community means of Amax did not show a clear elevational trend from 1000 to 3000 m (similar to the study of Fyllas et al. 2017), suggesting partial of full biochemical compensation of the temperature decrease, perhaps through a higher Rubisco concentration at higher elevation. However, Wittich et al. found a significant Amax decrease from 2000 to 3000 m, i.e. in the uppermost transect, which coincides with a significant decrease in foliar N and P concentrations in this transect section (which is only partly compensated by a LMA increase with elevation). In addition, leaf longevity increases toward higher elevation in this transect (Moser et al. 2007), which usually coincides with reduced foliar N and lower Amax. Both the Fyllas et al. and the Wittich et al. study suggest that temperature effects on the photosynthtic apparatus are mainly expressed through species turnover and related change in leaf traits. However, while the Wittch et al. data indicate only a small, and the Fyllas et al. data no, elevational change in Amax (measured at ambient temperatures), a pantropical literature review covering ca. 170 species from 18 sites (included in Wittich et al. 2012) suggests that Amax decreases by on average 1.3 µmol m$^{-2}$ s$^{-1}$ per km elevation in tropical mountains. Our conclusion is that local edaphic (soil fertility) and climatic factors (such as cloudiness or very high precipitation, causing temporal soil anoxia) may have a large influence on elevational patterns in phytosynthetic capacity. The Andes in Peru and Ecuador may differ from Central American, African or SE Asian tropical mountains in this respect. Our results do not allow more detailed conclusions on direct and/or indirect temperature effects on NPP, as temperature most likely influences various processes simultaneously, notably photosynthesis, respiration, stem growth, nutrient and water acquisition and morphology (temperature effects on wood and leaf properties).

**Reviewer#2:** However, this outcome (Wittich) appears to be inconsistent with the results presented by Fyllas et al 2017 who argue that Amax is constant with elevation and that this constancy is achieved through species turnover. In the Fyllas argument, the changes in species composition with elevation/ temperatures lead, on average, to a higher photosynthetic capacity at higher elevation (ie biochemical capacity, Vcmax25 – see Bahar et al. 2016 New Phytologist and van der Weg 2012 Oecologia for the data on Vcmax25), and whilst the temperature is lower, the effect on Amax (via higher Vcmax) is for Amax to be constant across elevations. For this reason, Fyllas et al argue that radiation and leaf photosynthetic traits drive productivity, not temperature (temperature is inferred to be an indirect driver of average trait variation). The data used by Fyllas are taken at leaf temperatures that do not go very cold even though we know leaves at high elevation spend much time well below 20 C (see van der Weg 2014), so the Wittich analysis might be of

much interest to the analysis presented in this paper from Ecuadoran sites, depending on how temperature is handled (remember that 24 hr (/12 hr!) photosynthesis is what determines overall assimilation totals not the maximum observed value of net photosynthesis, Amax, or the maximum photosynthetic capacity, Vcmax25) .

**Answer:** As the Wittich et al. data were taken at characteristic noon temperature conditions at the different elevations, we feel that they are relevant in the context of net primary production. We are aware that Amax data are poor estimates of daily canopy carbon gain. In another study on the carbon balance of the Ecuadorian forests along the elevation transect, we have roughly estimated canopy carbon gain by accounting for reduced photosynthesis due to cloudiness and lower temperatures in morning and evening hours (Leuschner et al. 2013).

**Reviewer#2:** Finally, there is a diversity question. The authors show that species diversity is very weakly related to WP in their data. The analysis also shows (and states) that variation in WP is large at each site (possibly larger than overall mean change in WP with elevation?...as also seen in Vcmax25 data, Bahar 2016, referred to above); and the argument is presented that this variation in WP reflects local variation in the environment/diversity. However, their discussion could be enhanced quite a bit by considering species composition and relations to diversity in mean traits (apols if this distinction is made and I've missed it). The Fyllas argument suggests that changes in mean traits do strongly drive variation in WP, and that the traits change with species turnover, ie species composition. Thus, it may be that raw species diversity does not explain variation in WP very well, but other elements of that diversity, ie mean trait change, conceivably might.

Overall, it seems that there is a key element missing here in synthesizing evidence for the mechanistic explanation of the observed variation in WP with elevation (ie, the argument that temperature drives the observed differences – ie, a/the principal focus of the paper). One part of it is explaining the different modelling approaches/conclusions and the other is relating this to evidence from the cited leaf physiology studies and the related mechanistic (/modelling) analysis, both in terms of the full variation in temperature at each elevation and in terms of the effect on diversity in traits (as well as species). I hope this extended comment is useful to the authors, as the data they have are fascinating and have the capacity to shed additional light on this overall discussion about a large fundamental question in forest ecology, and especially so given their additional nutrient information.

**Answer:** We agree that mean trait change is a major driver of change in productivity and other ecosystem processes along elevation gradients. This is shown by the recently published study on root traits along the Ecuador transect (Pierick et al. 2020, New Phytol) – the study also demonstrates the large effect of phylogeny on functional traits. However, the Amax data of Wittich et al. suggest that the NPP decrease is partly caused by a reduced photosynthetic gain, which contrasts with the constant Amax values in the Peru transect. We explain the differences by local edaphic and possibly climatic pecularities, and possibly by the effect of phylogeny, i.e. trait differences. This is expressed now in the Discussion. In addition, the Introduction was also partly rewritten to address this point.

**Reviewer#2:** Line 48. Should Tanner et al. 1998 be cited here in addition to the other place(s) where it is already?

**Answer:** Thank you. Was added.

**Reviewer#2:** Line 51. Should van der Weg 2014 be cited here in addition to the other place(s) where it is already?

**Answer:** Thank you. Was added.

**Reviewer#2:** Line 53, 69, elsewhere. It's not clear where mechanism and correlation are separated. This may be a timescale issue. The fertility discussion seems to be mainly by correlation but the impacts on photosynthesis need to be mechanistic; they are linked of course.

**Answer:** That was changed in the respective sentences and should be clearer now.

**Reviewer#2:** Line 80. The 'predestined' term might be replaced by '...which makes them attractive for the study of..'.

**Answer:** Thank you – was changed accordingly.

**Reviewer#2:** Line 89. The mention of assumed diversity effects is good, as we are not sure if they are real or if they only seem to occur in small plots. Perhaps this can refer to the preceding discussion as well, to tie up the idea of 'assumed diversity effects'?

**Answer:** The suggested diversity effects are critically introduced in the Introduction.

**Reviewer#2:** Line 262. Zimmermann 2010 (Glob Biogeochem Cycl) reports soil moisture data for a similar Andean elevation gradient that suggest little moisture limitation across elevations because of high rainfall, as here – it looks like it would be useful for you to cite here to substantiate your (reasonable) assumption.

**Answer:** We added the reference of Zimmermann et al.( 2010) to Introduction and Discussion.

**Reviewer#2:** Line 288. Need to refer to more modelling/data studies than just Fyllas here?

**Answer:** We added two other references (Finegan et al. 2015, Malhi et al. 2017).

**Reviewer#2:** Line 318. How does WSG mediate this effect on productivity? Can the authors suggest a process or mechanism? Leaving it open like this seems insufficient. Variation in WSG is often associated with variation in moisture constraints because of the link between WSG and hydraulic vulnerability (in some studies)…but the authors don't mention this. In this wet TMF environment, is it nutrients/growth rate, or even herbivory pressure, or just taxonomic identity that are related to the variance in WSG?

**Answer:** We added the following sentences here: A recent study in South-east Asian tropical forests showed that WSG does neither have a direct mechanistic effect on biomass production (Kotowska et al., 2020) nor on tree water consumption, even though harder wood tends to be associated with lower growth rates (Muller-Landau, 2004; Hoeber et al., 2014). Yet, WSG is known to be associated with most structural and functional wood properties (Chave et al., 2009), and it may also be related to anatomical attributes such as pit membrane characteristics that influence sap flux density, which itself is positively related to productivity (Kotowska et al., 2020).

**Reviewer#2:** Line 329. Can you shorten the text here by saying that the difficulty of LAI estimation may lead to both under-estimation where leaf clumping is dominant, or over-estimation where stem density is high?

**Answer:** We shortened the paragraph accordingly.

**Reviewer#2:** Line 349. Useful to refer here to the Sullivan 2016 work here on plot size/diversity/carbon storage.

**Answer:** We added the sentences: Similar to the DPR, a significant positive diversity – aboveground biomass relationship was observed in tropical lowland forests only in small plots (0.04 ha), while it disappeared at larger scale (1 ha) (Sullivan et al., 2017). Moreover this study showed that the diversity influence on AGB was small: Doubling species richness at 0.04 ha increased biomass only by 6.9%.

**Reviewer#2:** Line 368. The analysis here is clear in that it interprets temperature as a key driver of WP-elevation variance. But in the element where future needs are considered it could usefully also refer to a need to understand variation in overall traits – either trait diversity or change in mean trait values, and how they affect major ecosystem processes including productivity, in relation to temperature effects on the core driving processes themselves (ie as well as the species diversity question).

**Answer:** We added a sentence in the Conclusion referring to the need for more trait-related data.

**References:**

Hoeber, S., Leuschner, C., Köhler, L., Arias-Aguilar, D., Schuldt, B.(2014) The importance of hydraulic conductivity and wood density to growth performance in eight tree species from a tropical semi-dry climate. Forest Ecology and Management, 300, 126-136.

Kotowska, M., Link, R., Röll, A., Hertel, D., Hölscher, D., Waite, P.-A., Moser, G., Tjoa, A., Leuschner, C., and Schuldt, B. (2021) Effects of wood hydraulic properties on water use and productivity of tropical rainforest trees. Frontiers in Forests and Global Change, 3, 598759.

Leuschner C, Zach A, Moser G, Homeier J et al. (2013) The carbon balance of tropical mountain forests along an altitudinal transect. In: Bendix J. et al. (eds) Ecosystem Services, Biodiversity and Environmental Change in a Tropical Mountain Ecosystem of South Ecuador. Ecol. Stud. 221. Springer, Berlin, pp. 117-139.

Moser G, Hertel D, Leuschner C (2007) Altitudinal change of leaf area and leaf mass in tropical mountain forests -  a transect study in Ecuador and a pan-tropical meta-analysis. Ecosystems 10: 924-935.

Muller-Landau, H. C. (2004) Interspecific and inter-site variation in wood specific gravity of tropical trees. Biotropica 36, 20-36.

Bruijnzeel, L.A., Veneklaas, E.J. (1998) Climatic conditions and tropical montane forest productivity: the fog has not lifted yet. Ecology, 79, 3-9.

---

## Author Response (AR3)

**Response to Associate Editor**

**Associate Editor Decision: Publish subject to technical corrections** (22 Jan 2021) by Sara Vicca
Comments to the Author:
Dear authors,

Thank you for your revised manuscript, which I am pleased to accept for publication in Biogeosciences.
Before uploading your final files, please consider the minor remarks below.

Kind regards,
Sara

Minor remarks:
l. 119 wider C/N ratios: do you mean higher C/N ratios?
l. 120 higher SOM content does not necessarily reflect lower nutrient availability. In many cases, the opposite is true (see e.g. Van Sundert et al 2020, Global Change Biology, 26, 392-409). Please rephrase this sentence to make clear that the suggested negative relationship between SOM and nutrient availability applies to this specific study (presumably a temperature-effect?) but cannot be generalized.

Dear Associate Editor,

Thank you for accepting our manuscript for Biogeosciences!

We changed the respective sentence as follows.

"The increase in organic layer depth with elevation in both transects, accompanied by  greater C/N ratios and lower pH, is probably related to higher soil organic matter contents and indicates in our stuy areas a decrease in nutrient availability as a result of reduced organic matter turnover at lower temperatures."

Kind regards,

Jürgen